# Feeling Stressed and Ugly? Leave the City and Visit Nature! An Experiment on Self- and Other-Perceived Stress and Attractiveness Levels

**DOI:** 10.3390/ijerph17228519

**Published:** 2020-11-17

**Authors:** Claudia Menzel, Fiona Dennenmoser, Gerhard Reese

**Affiliations:** Social, Environmental, and Economic Psychology, Institute of Psychology, University of Koblenz-Landau, Fortstraße 7, 76829 Landau, Germany; fiona.dennenmoser@gmx.de (F.D.); reese@uni-landau.de (G.R.)

**Keywords:** attractiveness, stress, face perception, natural environment, urban environment

## Abstract

Natural environments, compared to urban environments, usually lead to reduced stress and positive body appreciation. We assumed that walks through nature and urban environments affect self- and other-perceived stress and attractiveness levels. Therefore, we collected questionnaire data and took photographs of male participants’ faces before and after they took walks. In a second step, female participants rated the photographs. As expected, participants felt more restored and attractive, and less stressed after they walked in nature compared to an urban environment. A significant interaction of environment (nature, urban) and time (pre, post) indicated that the men were rated by the women as being more stressed after the urban walk. Other-rated attractiveness levels, however, were similar for both walks and time points. In sum, we showed that the rather stressful experience of a short-term urban walk mirrors in the face of men and is detectable by women.

## 1. Introduction

To cope with everyday stress, stays and walks in natural environments have received increasing attention in research and media during the last decade (e.g., “forest bathing” or “shinrin-yoku”). In most scientific studies, the effects of nature walks are compared to walks in an urban environment. Thereby it is repeatedly found that the former has beneficial effects on mood, working memory, attention, self-reported restoration, brain activity, and one’s own body image [1,2,3,4,5,6]. Furthermore, self-reported stress and—albeit less consistently—physiological stress markers such as cortisol, heart rate variability, and blood pressure, are reported to be influenced positively during or after a stay in nature [4,7,8,9,10,11,12,13]. In sum, experience in natural environments (compared to urban environments) generally benefits human mental well-being and can alter physiological states.

As indicated above, recent studies have shown that contact with nature has beneficial effects on one’s own body appreciation [6,14,15]. These studies showed, on the one hand, an increase of participants’ satisfaction with their own physical appearance and a more positive body image after contact with nature. On the other hand, participants reported a decreased appreciation of their own appearance after stays in urban surroundings. Thus, short-term stays in (or virtual presentations of) different environments affect self-perceived attractiveness, in addition to the effects on stress and mood reported above. As discussed by Swami and colleagues [6,14], the elevation of body appreciation after an experience in nature may be due to a reduction of negative thoughts and affective states, as well as due to a shift in attention. These and the other beneficial effects of nature briefly reviewed above are in line with two major theories in the field, namely stress recovery theory (i.e., nature experience leads to more positive affective states and better stress recovery [16,17]) and attention restoration theory (i.e., nature exposure can restore directed attention [18]).

A different line of research indicates that stress over several weeks, its related markers, and/or its effects on health can be manifested in the face of a person: in a study using face photographs of students taken during either stressful exam times or less stressful periods, ratings by unfamiliar people on the perceived stress of the depicted person were different for the two periods of time ([19]; see also [20] for related findings on the detection of mental disorders and related markers, including happiness, depression, and distress, based on face photographs). Furthermore, it was found that stress markers, such as high levels of cortisol (often in an interaction with testosterone in men) and biomarkers of oxidative stress, negatively affects facial attractiveness ([21,22,23,24]; but see also [25,26,27]). Significantly, most of these studies used composite images to accentuate the respective markers, and/or the studies were cross-sectional and treated the hormone levels as a trait rather than a state factor. In fact, most studies investigating attractiveness focused on the innate characteristics of the face, such as prototypicality, symmetry, and having the respective sexual markers [28,29,30]. However, only few studies investigated the effects of situational or variable factors on attractiveness. Examples for the latter include the effects of skin color (e.g., melanin affected through sun exposure, beta-carotenoids influenced by diet, or redness evoked by increased blood flow [31,32,33,34]; but see also [29]).

Stress can be manifested in the face not only due to long-term effects (e.g., via hormone changes, see above; or air pollution [35]), but also due to short-term changes of the skin (e.g., altered blood flow [36,37]) or sleep deprivation [38], which affects facial attractiveness, too. In the current study, we bring together two lines of previous research and test the assumption that short-term experiences (i.e., walks) in nature and urban environments affect self- and other-perceived stress levels and attractiveness.

In sum, the reviewed literature shows that, first, nature compared to urban walks often has beneficial effects on mood, well-being, and stress. Sometimes, they also affect related physiological markers. Second, such markers and current moods can alter facial appearance, and may be detected by others. Third, nature walks affect body appreciation. However, it is yet unknown whether a short-term experience would also be mirrored in the face and would be recognizable by others. Therefore, the current study aimed at experimentally testing self- and other-reported stress and attractiveness after short-term nature and urban walks. Based on the literature, we expected that self-rated stress would be lower after a nature compared to an urban walk [4,8] and that the difference in stress levels after nature and urban walks would be mirrored in the face and detected by other raters [19]. Furthermore, we assumed that self-rated attractiveness would be higher after a nature compared to an urban walk [6] and that this difference would be reflected in the face and perceived by others [22,24]. We focused on men to avoid influences of changes in make-up, mood, self-perception, and facial appearance that might be associated with the menstrual cycle of women and can fluctuate within days [39,40,41,42]. The appearance of these men was then rated by unfamiliar women. 

## 2. Materials and Methods

The study consisted of two parts (Figure 1). First, male participants walked through nature and urban environments, while we took standardized photographs of their faces before and after each walk. Additionally, they completed questionnaires on their well-being, and self-perceived stress and attractiveness. Second, the photographs were rated for attractiveness and stress by female participants, who were unfamiliar with both the men and the design of the study. In the remainder of this article, we refer to the male participants as “walkers”, while the females are called “raters”. All subjects gave their informed consent before they participated in the study. The study was conducted in accordance with the Declaration of Helsinki and approved by the local ethics committee of the University of Koblenz-Landau, Germany (LEK-111). Before data collection, we pre-registered the study on AsPredicted.org (https://aspredicted.org/5eu33.pdf).

### 2.1. Participants

#### 2.1.1. Walkers

We recruited men by using email lists, social media, and direct contacts on the university campus. A total of 21 men aged 19 to 29 years (*M* = 23.62 years, *SD* = 2.84) participated in this first part of the study. All walkers were students and residents of the city in which the study was conducted, and the surrounding area at the time of the experiment. Twelve of the 21 walkers indicated doing nature walks regularly to cope with stress. Only three indicated using city walks. In addition, other often noted coping strategies were listening to music (16), meeting friends/family (16), watching TV/series/films (15), and physical activity (indoors (12) and outdoors (12)). Note that participants could select more than one option.

Due to technical reasons, the questionnaire data of one participant was incomplete (missing data for one time point) so that this participant was excluded from the analysis of this data. His photographs were, however, complete and we used them for the second study. All participants gave their informed consent before starting the experiment. For compensation, walkers could choose between partial course credit and €10. 

#### 2.1.2. Raters

In total, 84 women aged 18 to 35 years (*M* = 22.51, *SD* = 3.08) participated in the second part of this study. Most were students of the universities of Tübingen and Stuttgart-Hohenheim (Germany) at which the rating experiment took place (different from the university at which the first part of the study was conducted). They were recruited by email lists, social media, on-site advertising, and direct contact on the campus. All participants gave their informed consent before starting the experiment.

### 2.2. Procedure and Measures

The first part of the study (Figure 1) was conducted in April 2018, which was characterized by sunny weather and temperatures of 20–23 °C at the time of the walks. Walkers participated in small groups (see below) on two days in a row, completing the same procedure with the same group members on both days.

After arriving in the lab, each participant completed a questionnaire implemented in SoSci-Survey [43]. First, they rated their current state of attractiveness and stress using a continuous-looking scale (coded from 1 to 100) with “not attractive”/”attractive” and “restored”/”stressed” as endpoints. Second, they completed an adapted version of the Rest-Restless-Scale (aRRS; German equivalents to “I feel restless/placid/uneasy/relaxed/balanced/tense/nervous/calm”), which was based on the multidimensional affect rating scale [44,45], to indicate their current state of well-being. Third, individually each walker went to a neighboring room where a (female) photographer, who was naïve to the aim of the study, took a standardized photograph of his face (see below for details). Immediately after all pictures were taken (after about five minutes), the walkers were asked to follow the experimenter on a walk. The order of the walks was randomized between groups (i.e., ten walkers went to nature on the first day, eleven on the second day). Additionally, the walks were scheduled in a manner whereby the number of walkers and the time of walk (start of the session either 12 p.m. or 1 p.m.) was balanced and controlled for each environment and walker (i.e., quarter of walkers went to the city at the earlier time). Immediately after the walk, each walker individually went to the studio where another photograph was taken (the maximum was five minutes after the walk for the last participant of a given group). Then participants completed further questionnaires. First, the Restoration Outcome Scale (ROS; [46]) was used to measure perceived restoration. It was followed by the aRRS and the items on self-perceived attractiveness and stress. Both the aRRS and the ROS were used to confirm the intended effects of the walks. On the next day, the procedure was identical except that the participants walked in the other environment. Furthermore, the final questionnaire after the second walk included additional items on sociodemographics and general stress coping strategies. Overall, each session took about 50 min. In sum, self-reported well-being (measured by aRRS), stress, and attractiveness, as well as photographs of the face were obtained pre- and post-walk. Furthermore, after each walk restoration was surveyed using the ROS as a manipulation check. 

In the second part of the study, we asked female participants to spontaneously rate the attractiveness and stress of the person depicted on the photograph. All 84 photos collected in the first part (four pictures for each of the 21 walkers) were presented twice: for the rating of stress and attractiveness, respectively. The order of the two blocks (stress and attractiveness) was randomized across raters. Within each block, stimuli were presented individually in random order, while taking care that two or more images of the same walker did not appear in sequence. Below each picture, we presented a continuous-looking rating scale (coded from 1 to 100) that ranged from (German equivalents of) “stressed” to “restored”, or “unattractive” to “attractive”. The rating took about 10 to 15 min and was compensated with sweets. 

### 2.3. Walks

The walks were realized in small groups of three to five walkers accompanied by a female experimenter (FD). Each walker completed both walks on different days (see above). The routes were the same for all participants and started/ended at a building on the university campus where the photographs were taken and questionnaires were completed. The nature route led through a forest-like urban park close to the university (Figure 2A–C). The urban route was along both little-used and busy streets in the inner city area of Landau (Pfalz) in Germany, a town of about 45,000 inhabitants (Figure 2D–F). Walkers were asked to not talk to each other or to other people and to not use their smartphones. Both walks lasted about 22 min. The experimenter conducted all walks with constant speed and similar duration. For training, she went the routes several times before the beginning of the study.

### 2.4. Photographs

Before and after each walk, pictures of each walkers’ face were taken in portrait format (Figure 3A). Thus, in total 84 stimuli (four photos of each walker) were taken and then used for the second part of the study. Walkers were asked to show a neutral expression and to look directly into the camera. Their clothes were covered by a black scarf and were thus standardized to avoid differing reflections of the clothes on to the face. In addition, we asked walkers to not significantly change their hairstyles and beards for the two days of data collection. They sat on a stool in front of the camera. The camera was always at the same distance and was adjusted to the height of the respective walkers’ eyes. The lighting and camera settings were set manually and the same for all photographs. Pictures were taken with a Nikon D700 with an aperture of f/8, an exposure time of 1/100 s, a focal length of 105 mm, and an ISO of 200. Later, pictures were manually cut above the forehead and below the chin to remove the background (Figure 3B). Images were resized to 400 × 554 pixels. 

### 2.5. Data Analyses

Statistical analyses were conducted in R (version 3.4.3) using the interface RStudio (version 1.1.383, RStudio, PBC, Boston, MA, USA). Tests were two-tailed with an alpha of 0.05. For the aRRS and ROS, we formed sum scores (items with reversed polarity were inverted before). We compared ROS scores for the two environments using a paired *t*-test. The other comparisons were tested using a 2 (nature, city) × 2 (pre, post) repeated-measures ANOVA. The *p*-value for the interaction corresponds to the *p*-value of a paired *t*-test using difference scores (post minus pre values), which are often reported in the field [12,47]. All analyses, including the other-ratings by the female raters, were walker-based, meaning that the relevant *N* was always 21 (or 20 in case of the analyses with the missing data of one participant). Furthermore, we calculated Pearson correlations between stress and attractiveness ratings. 

## 3. Results

### 3.1. Self-Reported Measures by the Walkers

The ANOVA on the aRRS revealed neither a main effect of time (*F*[1,19] = 0.02, *p* = 0.887, η_p_^2^ < 0.01) nor walk (*F*[1,19] = 0.37, *p* = 0.552, η_p_^2^ = 0.02), but a significant interaction between the two (*F*[1,19] = 9.35, *p* = 0.006, η_p_^2^ = 0.33). Post-hoc tests indicated that participants felt more, yet non-significantly, at rest after a nature walk (*p* = 0.068; Table 1). The comparison of the ROS scores revealed that walkers felt more restored after a nature walk compared to an urban walk (*t*[19] = 3.18, *p* = 0.005, *d* = 0.61). 

The ANOVA on self-rated stress revealed neither a main effect of time (*F*[1,19] = 0.79, *p* = 0.385, η_p_^2^ = 0.04) nor walk (*F*[1,19] = 2.63, *p* = 0.121, η_p_^2^ = 0.12), but a significant interaction between the two (*F*[1,19] = 10.32, *p* = 0.005, η_p_^2^ = 0.35). Post-hoc tests showed that stress was lower after a nature walk (*p* = 0.026) and was lower compared to an urban walk (*p* = 0.029; Figure 4).

The ANOVA on self-rated attractiveness revealed neither a main effect of time (*F*[1,19] = 1.10, *p* = 0.307, η_p_^2^ = 0.05) nor walk (*F*[1,19] = 1.51, *p* = 0.235, η_p_^2^ = 0.07), but a significant interaction between the two (*F*[1,19] = 7.03, *p* = 0.016, η_p_^2^ = 0.27). Post-hoc tests showed that attractiveness ratings were higher after a nature walk (*p* = 0.022) and were higher compared to an urban walk (*p* = 0.041; Figure 4).

Stress and attractiveness ratings correlated negatively, but only for the ratings given before each walk (Table 1; see Appendix A for exact *p*-values).

### 3.2. Other-Rated Stress and Attractiveness

The ANOVA on other-rated stress levels revealed neither a main effect of time (*F*[1,20] = 0.45, *p* = 0.509, η_p_^2^ = 0.02) nor walk (*F*[1,20] = 0.79, *p* = 0.386, η_p_^2^ = 0.04), but a significant interaction between the two (*F*[1,20] = 9.79, *p* = 0.005, η_p_^2^ = 0.33; Table 1). Post-hoc tests indicated that stress was rated higher, yet non-significantly, after an urban walk (*p* = 0.058; Figure 4). 

The ANOVA on other-rated attractiveness revealed neither a main effect of time (*F*[1,20] = 0.26, *p* = 0.617, η_p_^2^ = 0.01), nor walk (*F*[1,20] < 0.01, *p* = 0.946, η_p_^2^ < 0.01), nor an interaction between the two (*F*[1,20] = 1.21, *p* = 0.285, η_p_^2^ = 0.06; Figure 4). 

Stress and attractiveness ratings correlated negatively, but not for the ratings given for the photographs that were taken after the urban walk (Table 1; see Appendix A for exact *p*-values). Self- and other-ratings of neither stress nor attractiveness correlated with each other (Appendix A). 

## 4. Discussion

This study’s aim was to examine whether short-time nature and urban experiences, implemented as walks, affect both self- and other-rated stress and attractiveness levels. In short, we replicated that nature compared to urban walks led to less self-reported stress, as well as to higher self-reported attractiveness [4,6]. Additionally, we showed for the first time that the reported stressful experiences from an urban walk are mirrored in the face of walkers and can be detected by unfamiliar people. However, these opposite-sex raters evaluated the participants equally attractive after both walks. In the following paragraphs, we discuss these findings in light of relevant literature and with regard to their implications for related research, public health, and society.

In line with previous research [8], the walkers reported significantly lower stress after the nature walk. Similarly, they felt more restored after the nature compared to the urban walk [4]. Moreover, reported feelings of rest point in a similar direction (measured by the aRRS). Therefore, we assumed that the walks had the intended effects on the walkers.

In addition to these manipulation checks, we found that nature walks positively affected the self-rated attractiveness of a person. This replicates previous findings [6,14,15,48,49], which showed that contact with nature leads to a more positive body image.

For the first time, we tested whether the experiences people made on short-term (i.e., 22 min) walks through nature and urban environments would be mirrored in their faces and detected by unfamiliar opposite-sex raters. The significant interaction between walk environment and the time point at which the photograph was taken revealed different effects of nature and urban walks on stress ratings by others. Note that this corresponds to a significant difference in difference scores, which were often the only reported analyses in the field [12,47]. Although our post-hoc tests were not significant, results indicate that walkers were perceived as being more stressed after they were in an urban environment compared to a natural environment. The mean values for each cell indicate that the differences in ratings were evoked by the experience in the urban rather than the natural environment. Thus, a relatively stressful walk in the city does not only affect how one feels but also how one is perceived by other people. Significantly, the effect found is of medium size (η_p_^2^ = 0.33 for the time × walk interaction, or Cohen’s *d* = 0.70 for the comparison of difference scores), despite a relatively small change in absolute numbers derived from the ratings. These relatively small but significant differences are most likely due to the design of the study. Since we presented each photograph independently and without subsequent occurrence of any pictures of the same walker, raters most likely applied their internal reference for ratings between the walkers and not within the four photographs of a single walker. Therefore, differences are expected to be relatively small. This approach is both more conservative and closer to real situations, compared to a design in which the four photographs are compared directly. A similar argumentation applies for the differences in evaluations between self- and other-rated stress and attractiveness: Walkers used their own affective states and feelings as a basis for ratings, while raters most likely used between-subject variation for evaluation.

Since self-rated attractiveness differed in dependence of the walk environment, and because certain stress markers may also be relevant for attractiveness perception (see Introduction), we expected that other-rated attractiveness would be also affected by the walk environment, albeit likely to a weaker extent. However, walkers were not rated as significantly more attractive after a nature compared to an urban walk. Thus, this finding does not extend previous results of a negative correlation between externally assessed attractiveness and the stress hormone cortisol [22,23,24]. However, in fact, cortisol is often not affected by short-term walks [11] and is discussed to reflect chronic rather than acute stress [8], which was likely the kind of stress that the raters in Little and colleagues’ study [19] detected in the faces of the students photographed in stressful exam times. To our knowledge, there are no studies investigating a relationship between more sensitive measures of short-term stress, such as heart rate variability or skin conductance, on facial appearance. Significantly, self-rated attractiveness differences may be also due to other psychological mechanisms that will not be mirrored in the face (e.g., more social encounters and thus comparisons with others in the city; shifts in attention). In fact, the absence of significant correlations between stress and attractiveness ratings for post-walk ratings indicates that the effect of the city may be stress-specific and represented by markers not relevant for attractiveness. Note that the aim of this study was to test whether short-term experiences are mirrored in the face. Future studies are needed to determine and disentangle the underlying mechanisms (i.e., cortisol, blood flow, expression etc.) of our findings. The first evidence comes from two recent studies from China. In one study, selfies uploaded in social media were related to the location at which the picture was taken [50]. This study indicated that software-measured facial expressions, especially from women, vary with the distance to the city center. A related study [51] suggests that the characteristics of urban forest environments (e.g., greenness) are associated with positive facial expressions, which were recorded more often in such forest environments compared to built-up environments. These studies suggest a relationship between facial expressions and the environment, but more research is needed to disentangle the effects of experiences in natural versus built-up surroundings on facial appearance.

In informal feedback, some raters indicated that they found it easier to give independent ratings on stress rather than attractiveness because the latter was perceived to be a situation-independent state, whereas stress was perceived naturally depending on the situation and time. Perhaps the evaluation of a factor that correlates strongly with attractiveness but is perceived as more variable, such as health, would lead to greater variations in ratings. However, other studies that showed slightly different images of the same person still found variance and effects in attractiveness ratings [52,53]. Some of our results might be non-significant due to low power. However, given the complexity and effort of this study, realizing such an experiment with a sample size based on a standard a-priori power analysis seems unrealistic (e.g., assuming a small effect of η^2^ = 0.02 or *f* = 0.14 with a power of 0.8 for a 2 × 2 interaction, 388 walkers would be necessary). We nonetheless suggest that future research seeks to increase walker numbers in a feasible way, and to include all genders.

Further investigations are needed to show how the short-term effect of restoration or stress on a person’s face may become visible and affect the ratings of others. Possible mechanisms might be emotional expressions that differ between a relaxed and tense state. By instructing our walkers to show a neutral expression when being photographed, we might have obscured this mechanism and effect [54]. However, we deliberately decided against encouraging participants to show their emotions to have a more controlled stimulus set and to avoid overwriting of other markers due to the expression. Nonetheless, despite our instruction for a neutral expression, the facial expression still might be different between the conditions because participants unconsciously might have different muscle activity and thus expressions depending on their mood and stress [55,56,57,58,59]. These small differences might have led to a difference in other-perceived stress ratings. Furthermore, differences in stressors, such as air quality and noise, and/or emotional states might have led to different skin conditions and/or levels of blood flow in the face, which might be detected by the raters [60,61,62,63]. 

Based on the current state of research and available data, we cannot conclude on the relationship between short-term nature experiences and other-perceived attractiveness. In fact, the current study has one major limitation, namely the relatively small sample size of walkers, and the focus on one gender. However, the sample size is comparable to previous studies with similar designs [10,13,19,38]. Moreover, the study and the photographs were highly standardized so that noise in the data was reduced to a minimum. Another reason for the lack of changes in attractiveness might be that in this study original photographs of the individuals were used for ratings. Most studies reporting a relationship between cortisol and attractiveness used composite images that were created by morphing together face images of several individuals (but see [23]). Other promising approaches might be to provide more information to the raters, e.g., by presenting videos of the walkers, which include spontaneous expressions and body movements. 

## 5. Relevance

The current study shows that self- and other-rated stress levels, as well as self-rated attractiveness, are negatively affected by short-term urban experiences. Thus, this work not only contributes to the research areas of restorative environments and face perception but also provides important insights for public health. First, the study adds support to the general finding of nature (compared to cities) as a health-promoting environment and nature walks as an intervention strategy to reduce stress. Second, the current study shows that urban environments can be stressful and affect our appearance. Note that current data were collected in a rather small town. Larger or even mega-cities might yield much more pronounced effects. Third, many (especially younger) people struggle with their own body, and nature experiences seem to be a promising intervention strategy to reduce body- and attractiveness-related suffering [6,14]. Fourth, stress can worsen the perceived own body image and, vice versa [64,65], perhaps lead to reinforcement effects. Fifth, stress can be contagious [66,67], and detecting stress from photographs might extend the contagious effect to a broader range of people, for example via social media such as Instagram. Finally, own appearance can be a motivator for healthier behaviors, such as nature walks as a coping strategy against stress [68]. In sum, nature walks seem to be an effective, accessible, and easy stress intervention to buffer the stressors of urban environments and to attenuate their negative effects on society.

## 6. Conclusions

With the current study, we showed that walking experiences in natural environments has beneficial effects on subjective restoration, stress, and attractiveness levels. Moreover, for the first time, we tested whether these subjective changes are mirrored in the face. Indeed, our data suggests that the relatively stressful experience in urban environments, compared to natural environments, manifests in the face and can be detected by others.

## Figures and Tables

**Figure 1 ijerph-17-08519-f001:**
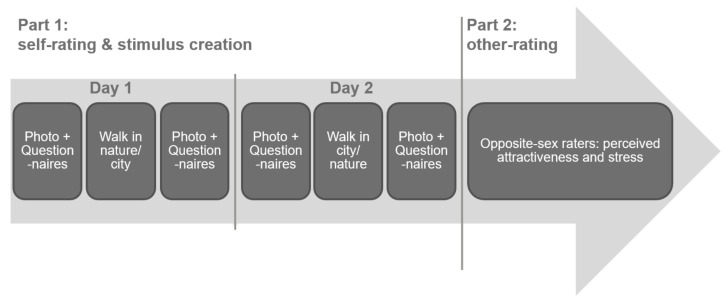
Overview of study design and procedure.

**Figure 2 ijerph-17-08519-f002:**
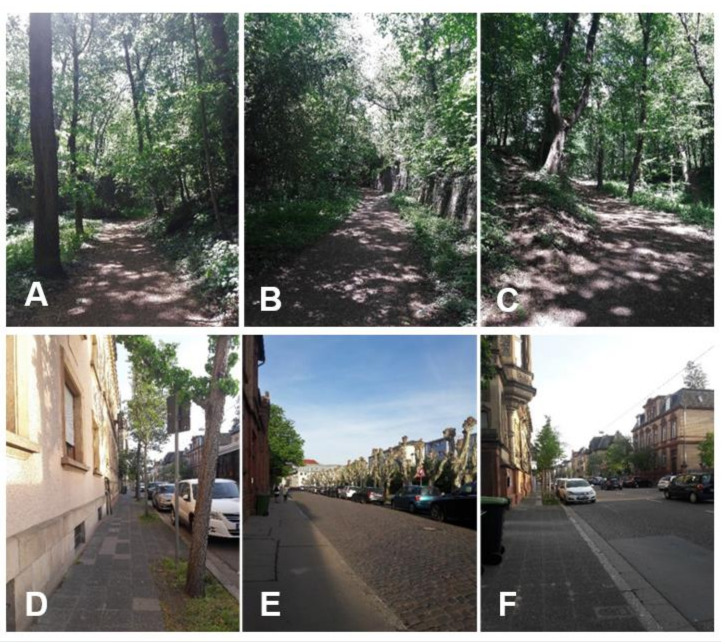
Representative images of the walking routes through nature (**A**–**C**) and urban (**D**–**F**) environments. © The authors.

**Figure 3 ijerph-17-08519-f003:**
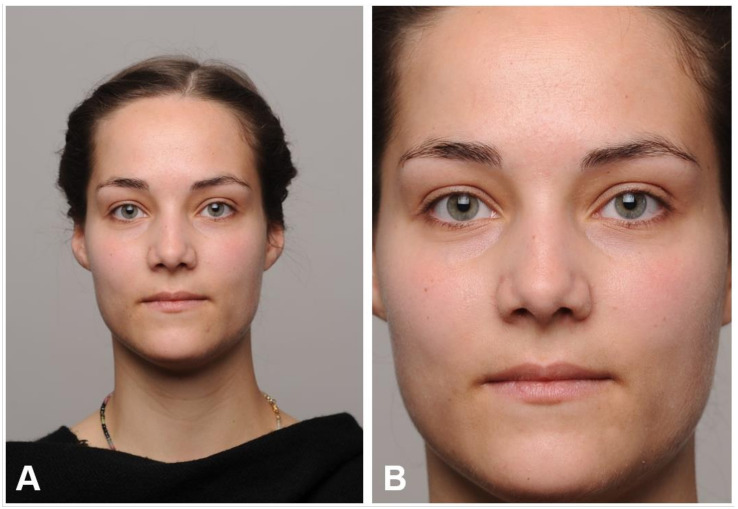
Example picture to represent photographs taken of each walker in an original (**A**) and cropped version (**B**). The latter was used for the ratings. For reasons of data protection, no photograph of a walker is shown here, but of one of the authors (FD) agreed to have her picture published here. Cropping, lighting, and camera settings correspond to those of the walkers. © The authors.

**Figure 4 ijerph-17-08519-f004:**
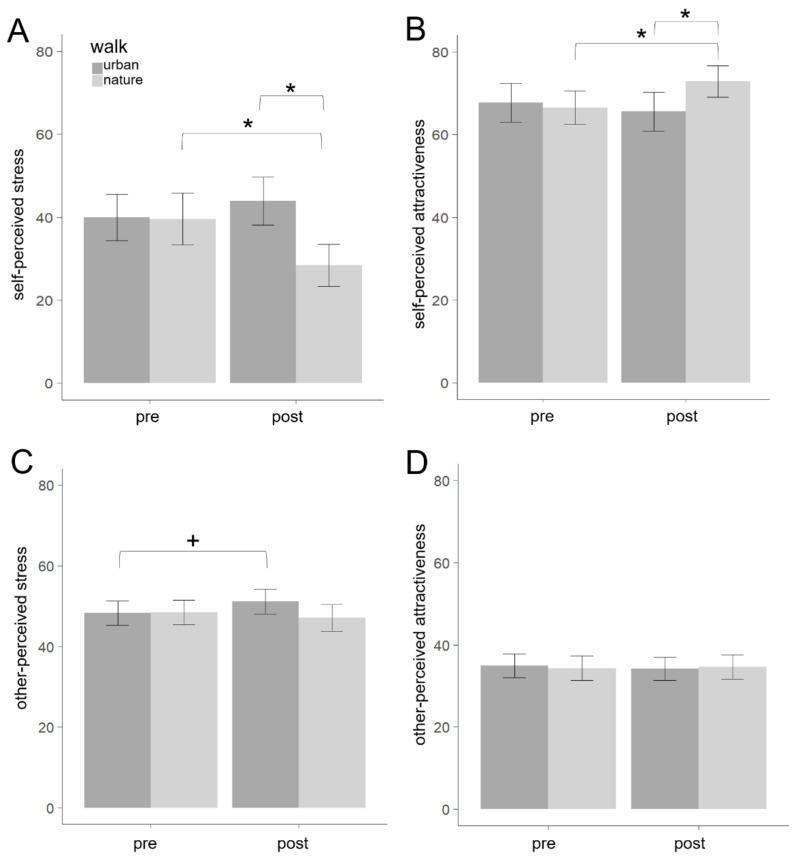
Results of self- (**A**,**B**) and other-rated (**C**,**D**) stress and attractiveness before and after urban and nature walks with indication of significance for post-hoc tests. See main text for statistical parameters of the main effects and interactions from the respective 2 × 2 ANOVAs. Note. * *p* < 0.05, + *p* = 0.058.

**Table 1 ijerph-17-08519-t001:** Mean ± *SD* of dependent variables, as well as correlation between stress and attractiveness ratings.

Rater	Self				Other			
Time	Pre		Post		Pre		Post	
Walk	Nature	Urban	Nature	Urban	Nature	Urban	Nature	Urban
aRRS	27.95 ± 7.53	29.15 ± 5.19	30.10 ± 4.94	27.30 ± 6.69	n.a.	n.a.	n.a.	n.a.
ROS	n.a.	n.a.	25.35 ± 4.28	21.70 ± 4.33	n.a.	n.a.	n.a.	n.a.
stress	39.60 ± 27.93	39.95 ± 25.00	28.40 ± 22.75	43.90 ± 25.71	48.43 ± 3.07	48.30 ± 3.00	47.09 ± 3.33	51.11 ± 3.11
attractiveness	66.50 ± 18.20	67.70 ± 21.13	72.85 ± 17.13	65.55 ± 21.22	34.33 ± 2.99	34.90 ± 2.89	34.63 ± 2.94	34.19 ± 2.80
correlation	−0.73 ***	−0.54 *	−0.34	−0.18	−0.63 **	−0.61 **	−0.60 **	−0.17

*Note*. *** *p* < 0.001, ** *p* < 0.01, * *p* < 0.05, n.a. = not available.

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
