# Peer review of "Feeling Stressed and Ugly? Leave the City and Visit Nature! An Experiment on Self- and Other-Perceived Stress and Attractiveness Levels"

_ijerph, 2020, doi:10.3390/ijerph17228519_

Round 1

Reviewer 1 Report

I have attached my comments in a word doc for suggestions etc, but just to note I really enjoyed reading this. I hope my comments help to improve the MS and that it's out in the world shortly! 

Author Response

We uploaded a word file with our responses.

Reviewer 2 Report

Contact with nature is an important health resource. This article expands the understanding of these relationships by examining the facial expressions of people,

thus contributing to a deeper understanding of human-environment relations.

The article is written fluently and logically structured.

The article is very descriptive a theoretical underpinning is missing.

Why does contact to nature lead to a positive body image?

Several studies already focus on facial expressions of forest visitors and environmental factors:

e.g. We et al. 2019: Facial Expressions of Visitors in Forests along the Urbanization Gradient: What Can We Learn from Selfies on Social Networking Services?

these studies should be taken into account

Author Response

(The authors gave the same response as above.)

Reviewer 3 Report

Excellent paper, very thorough introduction and background, well-designed and explained study methods, discussion and conclusions match the data.  

Only suggestion is to possibly include a graph of the data (rather than just Table 1), to enhance the presentation and shareability of the data.

Author Response

(The authors gave the same response as above.)

Reviewer 4 Report

This is an unique article of investigating whether walking in nature and relieving stress makes people's faces more attractive. But research methods need to be reviewed.

  1. There was little change in other-rated stress and attractiveness in nature. There has been no proper discussion of this point.

  1. The results of other-rated evaluations resulted in much higher stress and much less attractiveness than self-rated evaluations. Could this be due to the photo giving such a bad impression than the real face? Photographs only express a static expression, so author should consider whether it really represents a person's appeal. Using video or other facial expressions, or body movements, might be more reflective of this.

  1. It is recommended to plan the experiment that the number of steps and physical activity would be the same when walking in nature and in the urban.

  1. Author should make sure that whether the walkers applied cosmetics when taking the photos even they were men.

Author Response

(The authors gave the same response as above.)

Round 2

Reviewer 4 Report

The author responded politely to the reviewer and corrected the manuscript appropriately.